# Revisiting the Function of p21^CDKN1A^ in DNA Repair: The Influence of Protein Interactions and Stability

**DOI:** 10.3390/ijms23137058

**Published:** 2022-06-24

**Authors:** Giulio Ticli, Ornella Cazzalini, Lucia A. Stivala, Ennio Prosperi

**Affiliations:** 1Istituto di Genetica Molecolare “Luigi Luca Cavalli-Sforza”, Consiglio Nazionale delle Ricerche (CNR), Via Abbiategrasso 207, 27100 Pavia, Italy; giulio.ticli@igm.cnr.it; 2Dipartimento di Biologia e Biotecnologie, Università di Pavia, Via Ferrata 9, 27100 Pavia, Italy; 3Dipartimento di Medicina Molecolare, Università di Pavia, Via Ferrata 9, 27100 Pavia, Italy; ornella.cazzalini@unipv.it (O.C.); luciaanna.stivala@unipv.it (L.A.S.)

**Keywords:** p21^CDKN1A^, DNA repair, DNA damage response, PCNA, protein degradation

## Abstract

The p21^CDKN1A^ protein is an important player in the maintenance of genome stability through its function as a cyclin-dependent kinase inhibitor, leading to cell-cycle arrest after genotoxic damage. In the DNA damage response, p21 interacts with specific proteins to integrate cell-cycle arrest with processes such as transcription, apoptosis, DNA repair, and cell motility. By associating with Proliferating Cell Nuclear Antigen (PCNA), the master of DNA replication, p21 is able to inhibit DNA synthesis. However, to avoid conflicts with this process, p21 protein levels are finely regulated by pathways of proteasomal degradation during the S phase, and in all the phases of the cell cycle, after DNA damage. Several lines of evidence have indicated that p21 is required for the efficient repair of different types of genotoxic lesions and, more recently, that p21 regulates DNA replication fork speed. Therefore, whether p21 is an inhibitor, or rather a regulator, of DNA replication and repair needs to be re-evaluated in light of these findings. In this review, we will discuss the lines of evidence describing how p21 is involved in DNA repair and will focus on the influence of protein interactions and p21 stability on the efficiency of DNA repair mechanisms.

## 1. Introduction

### 1.1. p21^CDKN1A^, a Multifaceted Protein

The p21^CDKN1A^ protein (also named p21^WAF1/CIP1/SDI1^), is a cyclin-dependent kinase (CDK) inhibitor universally known to be an important player in the maintenance of genome stability, in particular after genotoxic damage [1,2,3].

The main function performed by p21 as a CDK inhibitor is to block cell-cycle progression, not only in response to DNA damage, but also in other more physiological processes requiring cell-cycle arrest, such as the quiescence phase (G_0_), senescence, and differentiation [4,5]. Cells with the deletion of the p21 gene show a premature S-phase entry, with important consequences on genome stability, both in the absence and in the presence of DNA damage [6,7]. These lines of evidence have highlighted the tumor suppressor properties of p21 as a regulator of cell cycle progression, not only in the G_1_ and G_2_ phases of the cell cycle but also in mitosis [8]. In addition, more recent results have proposed a new function of p21 during the S phase, i.e., the fork-speed regulation in the elongation phase of DNA replication [9,10].

In addition to the cell cycle, p21 is involved in other important pathways, including transcription, apoptosis, DNA repair, and, more recently discovered, cell motility [3,11]. The participation of p21 in these pathways occurs not only in response to DNA damage but also in physiological conditions, such as the maintenance of tissue homeostasis by regulating stem cell proliferation [3,11]. The function of p21 in such pathways is as a regulator/inhibitor, occurring via specific interactions with relevant proteins participating in these cell programs. In fact, structural studies have revealed that p21 is relatively disordered in the free state, a feature enabling the binding to a variety of partner proteins [12]. Being a small protein (18.5 kDa), short amino acidic stretches located in the N- or C-terminal region of the protein confer specific binding properties and regulatory activities, such as the CDK-binding domain and the PCNA-interacting protein (PIP) box (Figure 1).

These multiple functions of p21 have been treated extensively elsewhere [3,11], and will not be discussed further here.

Although p21 is predominantly considered a tumor suppressor, it can also exhibit oncogenic properties in certain conditions, e.g., by inhibiting drug-induced apoptosis in tumor cells, and/or promoting metastatic potential through cell motility [13]. The dual behavior of p21 has been referred to as a double-edged sword, hence suggesting the definition of a “two-faced genome guardian” [14]. However, its negative side is exhibited solely in tumor cells, in which it might circumvent the canonical cell response to DNA damage by redirecting p21 functions towards pathways facilitating proliferation and/or avoiding cell death [13]. 

At the basis of these behaviors are mechanisms (e.g., phosphorylation) allowing p21 to shift localization from the nucleus to the cytoplasm, where it can inhibit apoptosis [15], thereby facilitating the escape of tumor cells from drug-induced cell death [16,17]. Another example is provided by p53-deficient cell model systems in which it has been shown that a subset of cells may acquire the ability to evade senescence after the prolonged expression of p21, resuming proliferation at the expense of DNA replication stress [18]. 

These examples are very useful to understand the double-face of p21, yet they have generated some confusion about p21’s physiological functions, especially when these aspects have been investigated in tumor cell lines without a comparison with non-transformed cell model systems. While this approach may have facilitated the studies, the results thus obtained have complicated the picture because transformed cell lines may have genetic defects, influencing, directly or indirectly, the function of various cellular processes, including DNA repair [19,20]. Probably for these reasons, the latter is the typical field of investigation in which the role of p21 is still confused and debated [21]. 

### 1.2. p21 and DNA Repair 

The role of p21 in DNA repair was initially investigated due to its ability to inhibit DNA synthesis [22,23] through the high-affinity binding to PCNA, the master regulator of DNA replication [24]. PCNA is also involved in almost all known mechanisms of DNA repair requiring new synthesis to restore the original nucleotide sequence. These processes include nucleotide excision repair (NER), base excision repair (BER), mismatch repair (MMR), as well as homologous recombination (HR) and non-homologous end-joining (NHEJ) [25,26,27,28]. Therefore, whether p21 could inhibit DNA repair became a question to address.

The greater binding affinity of p21 with PCNA leads to the displacement of other PCNA-interacting proteins by disrupting their association with PCNA [24]. However, both in vitro and in vivo studies investigating the effect of p21/PCNA interactions on DNA repair provided different and contrasting results, which are complicated by the variety of model systems and p21 expression conditions [2,21]. 

In this review, we will discuss the lines of evidence for or against the participation of p21 in various processes of DNA repair, focusing on the influence therein of p21 protein stability and interactions, in relation to the functional response to DNA damage, and the cell models investigated.

## 2. Influence of p21 in DNA Repair Systems: Protein Interactions

In the following, we will analyze the studies reporting results on the influence (positive, negative, or null) of p21 interactions with proteins participating in various mechanisms of DNA repair. In fact, in addition to PCNA, p21 may directly bind to other proteins directly or indirectly involved in DNA repair, further supporting the guardian functions of p21 in the pathways of genome integrity maintenance. A roadmap of the discovery of these interactions is reported in Table 1.

### 2.1. Direct Reversal

Direct reversal is a DNA repair process enabling the removal of alkyl adducts to DNA by the activity of factors, without requiring new DNA synthesis. One of the most important enzymes performing such a reaction is O^6^-methylguanine DNA methyl transferase (MGMT), which has been shown to interact with both PCNA and p21 [29]. Evidence indicated that the absence of p21 renders cells more sensitive to alkylating agents, resulting in a greater co-localization of MGMT with PCNA. In such conditions of absent or reduced p21 protein levels, cells were shown to be less proficient than their p21-expressing counterparts to remove the lesions, indicating a positive effect of p21 on MGMT-induced DNA repair. A mechanism based on a transcriptional effect on MGMT expression induced by p21 was found. However, it was also shown that MGMT was downregulated by degradation with a mechanism dependent on a member of the Cullin 4 Ring E3 ubiquitin ligases (CRL4) family, namely CRL4^Cdt2^, and requiring the interaction of MGMT with PCNA [29]. Interestingly, although MGMT did not interact directly with p21, the existence of ternary complexes containing the three proteins suggested that p21 could displace MGMT from PCNA. From this point of view, higher levels of p21 relieved the interaction of MGMT with the PCNA/CRL4 complex, thereby explaining the greater DNA repair activity shown by p21-expressing cells vs. those lacking the protein [29]. 

### 2.2. Nucleotide Excision Repair (NER)

The NER system is one of the most characterized DNA repair processes because of its ability to remove helix-distorting lesions and bulky adducts, such as UV-induced cyclobutane pyrimidine dimers (CPDs) and 6,4-photoproducts (6,4-PPs). NER is characterized by two sub-pathways, one associated with transcription, or transcription-coupled repair (TCR), and the other occurring in the rest of the genome, named global-genome repair (GGR) [30]. NER has been the first, among the DNA repair processes, in which the role of p21 was studied, both in vitro and in cultured cells, in relation to the binding to PCNA. Several biochemical studies investigated the influence of p21 in reconstituted in vitro assays, using, as exogenous substrates, plasmids or artificial double-stranded DNA fragments, either UV-irradiated or containing a cisplatin adduct. DNA repair factors were provided by cell extracts and/or by purified proteins, while a recombinant p21 protein (His- or GST-tagged) was used. The first studies found a different effect of p21 on DNA replication vs. DNA repair, i.e., p21 inhibited only the first process [31,32]. The length of the DNA synthesized (long vs. short patches) was indicated as the possible mechanism responsible for the different sensitivity of the two processes to p21. Other studies provided opposite results with inhibitory effects, probably depending on the high concentrations of p21 used in the assays [33,34]. However, discrepancies in the results were also attributed to other reasons, such as repair substrates, recombinant p21, or the nature of the repair factors used in the reactions [21]. Interestingly, the NER resistance to p21 was further characterized in a study using a UV-irradiated plasmid: the results indicated that repair inhibition by p21 could be observed only by uncoupling the incision from the DNA synthesis step [35]. A particular warning was the verification that in cell extracts, the synthesis step was actually dependent on NER and not, for instance, on the repair of base damage or nick translation [35]. NER inhibition was observed with a C-terminal (p21C) but not with an N-terminal peptide in a reaction using cell extracts and a UV-damaged plasmid [36]. Interestingly, at a very low concentration (50 nM) of p21C, an increase in the repair assay was observed, which was, however, reversed by increasing the concentration, up to a p21/PCNA ratio of 50:1. Increasing the amount of PCNA restored the repair activity and no inhibition was observed at a 0.5:1 ratio. The inhibitory effect of the p21C fragment was also found in human fibroblasts (HFs) using electroporation or a permeabilization procedure to introduce the p21 fragments into living cells [36]. 

The effect of p21 on NER was also investigated in vivo using cell model systems in which p21 was ablated, such as the HCT116/p21^−/−^ [37,38] or the p53-deficient DLD1 cell line, in which p21 expression (virtually absent) was driven by a Tet-regulated expression plasmid [39]. In both model systems, the repair of a damaged plasmid reporter, i.e., the host-cell reactivation (HCR) assay, analyzed in the presence or in the absence of p21, showed that p21 facilitated the repair of UV- or cisplatin-damaged DNA [37,38,39]. The dependence of these results on the interaction of p21–PCNA was demonstrated by the reversed effect when a truncated p21 peptide, lacking the C-terminus and thus unable to interact with PCNA, was reintroduced into HCT116/p21^−/−^ cells [37]. A similar positive role of p21 in the DNA repair of a reporter plasmid treated with the chemotherapy agent BCNU, a nitrosourea derivative shown to induce cross-links, was also found in glioblastoma cells expressing p21 under an inducible system [40]. The repair of UV-induced CPDs and 6,4-PPs was analyzed at the GGR level by monitoring the disappearance of these lesions with an immunoblotting assay, while TCR was analyzed in the transcribed strand of the DHFR or TP53 gene by Southern blot. In contrast with previous results, this study reported that both sub-pathways were not significantly influenced by the lack of p21 in HCT116/p21^−/−^ cells [41,42], although the HCR assay showed a lower efficiency in the p21^−/−^ vs. the p21^+/+^ cells [42]. Similar results were obtained with primary cultures of human lung fibroblasts (LF1), in which the p21 gene was disrupted by homologous recombination [43]. A small reduction or no significant differences between p21^−/−^ and parental p21^+/+^ cells were found in the GGR and TCR repair of UV lesion assays [41]. Remarkably, the same primary LF1 fibroblasts were used in another study, which reported the significant inhibition of CPD removal in p21^−/−^ vs. p21^+/+^ cells, both at the GGR and TCR levels [44]. The reason for this discrepancy is unclear and could be related to the nature of primary cultures and the different assay procedures. 

It is worth considering that the lack of differences in DNA repair efficiency does not mean that p21 is an inhibitor, as it was implied by another study in which TCR efficiency in DLD1 parental cells was compared with a cell clone carrying an ablation of the p21 gene (DLD1/p21^−/−^) [20]. In this study, however, the parental p21^+/+^ cells did not express detectable amounts of p21 because of p53 deficiency, i.e., a genetic background known to affect NER efficiency [45,46,47]. Therefore, the evidence that DLD1/p21^−/−^ cells repaired UVB-induced lesions more efficiently than parental cells should be analyzed in the context of p53 and NER deficiency [47]. 

Using rodent cell models, a relatively small reduction in NER efficiency (CPD removal assessed by immunoblot) was observed in p21^−/−^ murine embryonic fibroblasts (MEFs), as compared with p21^+/+^ cells, [48], and in p21^−/−^ murine keratinocytes [49]. A significant reduction was instead found in *gadd45/p21* double-knockout (ko) MEFs [48], while in the double-ko keratinocytes the DNA repair efficiency was restored, as compared with p21^+/+^ cells [50]. This discrepancy was explained by differences in proliferative ability and in the p21 levels of the two cell types [50]. The restoration of NER efficiency evaluated by UDS (autoradiography) was also observed in double-ko *DDB2/p21*, as compared with *DDB2* ko MEFs [51]. However, the double ko showed significantly higher levels of proliferating cells, thus making it difficult to identify true UDS from very early S-phase autoradiography signals [52]. In fact, double-ko *DDB2/p21* mice showed an accelerated tumor onset, and the restoration of NER efficiency did not protect against UV-induced skin carcinogenesis [52].

In HeLa cells, the expression of p21-GFP at levels inducing cell-cycle arrest in the G1 phase did not produce any inhibition of NER efficiency (evaluated with the UDS assay). Interestingly, the p21-GFP protein was shown to accumulate at DNA damage sites, depending on the interaction with PCNA and on the NER activation, since the recruitment of PCNA and p21 was significantly delayed in NER-deficient XP-A fibroblasts [53]. In normal fibroblasts, evidence was also provided that p21 could be immunoprecipitated in a complex containing both PCNA and the p125 subunit of DNA polymerase (pol) δ, thus implying that p21 did not disrupt their association [53]. In agreement with these findings, the expression of p21^wt^ or mutants lacking either CDK- or PCNA-binding domains did not affect any step of the NER process, while retaining the ability to inhibit DNA replication in U2OS cells [54]. 

The role of the N- and C-terminal p21 domains on the sensitivity to cisplatin and DNA repair was investigated in bladder cancer cells by CRISPR genome editing, mimicking clinically relevant truncated mutations. The loss of p21 (mutation truncating at the very N-terminus) reduced the DNA repair ability of DNA–platinum adducts, while the expression of a peptide containing the CDK-binding domain promoted cell resistance to cisplatin [55].

Following the discovery of an interaction of PCNA with the lysine acetyl transferase (KAT) p300 [56], whose activity may be regulated by p21 [57], the association of this enzyme with p21 was investigated and found to be involved also in NER. In particular, p21 relieved the inhibition of KAT activity by disrupting the association between p300 and PCNA, thereby restoring the acetylation levels of histone H3 [58]. The role of p300 was subsequently confirmed by the reduced efficiency of NER observed after siRNA-mediated depletion [59,60] and by the evidence that other NER factors, such as XPG, are acetylated by p300 [61], thereby regulating the chromatin association of XPG after DNA damage [59]. Interestingly, in p21^−/−^ HFs, chromatin-bound XPG was increased, in a similar manner as after the inhibition of XPG acetylation, suggesting that p21 modulates p300 KAT activity in NER not only at the histone acetylation level [59,61].

### 2.3. Base Excision Repair (BER)

BER is the mechanism devoted to removing damage, such as alkylation and oxidative lesions, at the level of single DNA bases. The process initiates with the lesion recognition by DNA glycosylases and processing by apurinic/apyrimidinic (AP) endonucleases, followed by DNA polymerases (DNA pols) catalyzing the synthesis of short or long patches and by DNA ligases involved in the final ligation step [62].

The participation of p21 in BER was initially investigated in vitro, where the inhibition of DNA pol δ activity in the long-patch BER was documented. However, in the presence of the APE1 endonuclease, BER efficiency was significantly restored, thereby attributing a regulatory role to this enzyme [63]. This finding thus indicates that BER efficiency in certain conditions is not inhibited by p21. Interestingly, it was later discovered that APE1 interacts with PCNA [64].

Another example of the p21-mediated inhibition of long-patch BER was provided in vitro using cell extracts of mouse embryonic fibroblasts from wt and DNA pol β-ko mice, after long-term treatment with plumbagin, an agent inducing oxidative DNA damage [65]. Using these cell extracts, an inhibitory effect of p21 was observed in reconstituted repair reactions with exogenous substrates in the presence of significant amounts of recombinant p21 protein, while the repair of DNA damage induced by plumbagin was not assessed [65].

The participation of p21 in BER was further supported by the direct interaction with poly(ADP-ribose) polymerase 1 (PARP1), another important BER player, which occurred independently of PCNA, both in vitro and in HFs treated with the alkylating agent MMNG [66]. In vitro, both p21 and PCNA inhibited PARP1 activity [66]. A positive influence of p21 in BER was found in p21^−/−^ HFs, which were more sensitive to DNA damage induced by alkylating agents and accumulated more DNA lesions than the parental p21^+/+^ cells, as evaluated by the Comet assay [67]. The absence of p21 resulted in the chromatin accumulation of PARP1, as well as XRCC1, DNA pol β, and PCNA itself, in concomitance with an excessive persistence of poly(ADP-ribose) (PAR) molecules [67], which could explain the reduction in BER efficiency [68,69]. 

Interestingly, in a mouse model of cigarette smoke (CS)-induced lung disease, a similar increase in PAR synthesis was found in p21^−/−^, as compared with wt mice, in concomitance with an increase in the total amount of Ku70/Ku80 [70]. However, this effect was interpreted as an increase in DNA repair, suggesting that the presence of p21 was detrimental to the process. The absence of p21 was also considered beneficial because of the reduced senescence consequent from CS exposure, although senescence is, in principle, a mechanism of cell-cycle exit, safeguarding genome stability [3].

Another study investigated the effect of an increase in the p21 protein induced by resveratrol in a model of MCF cells transformed by continuous exposure to CS condensate. In this model, long-term exposure to resveratrol induced DNA damage independently of the presence or the absence of p21 (siRNA-mediated depletion). However, a reduced DNA repair capacity of a reporter plasmid via the long-patch but not via the short-patch BER was observed in the absence of p21 [71]. In these long-term experiments, the depletion of p21 reduced the apoptotic response, while it increased the expression of some BER proteins. The increase in p21 protein levels induced by resveratrol occurred in concomitance with a reduction in the PCNA/FEN1 interaction, although an evaluation of the repair of resveratrol-induced DNA damage was not performed under these conditions [71]. A more recent study showed that the PARP1 binding to BER intermediates in vitro was stimulated by p21, whose association with PARP1 required PAR synthesis [72]. 

These results indicate a regulatory role of p21 in limiting PARP1 activity, in order to avoid an excess of PAR molecules, which may become toxic for the cells [68,69].

### 2.4. Non-Homologous End Joining (NHEJ)

Evidence from several studies indicates that p21 is also involved in DNA single- and double-strand break (DSB) repair pathways, such as non-homologous DNA end-joining (NHEJ) and homologous recombination (HR). p21 was first co-precipitated with Ku70, a central actor of NHEJ, after the γ-irradiation of peripheral blood lymphocytes. This association appeared as a transient phenomenon, being time- and dose-dependent (detectable within 2 h after irradiation, at a 2 Gy dose but not at 4 Gy). The authors suggested that the binding of Ku70 to p21 was involved in deciding the cell fate in response to irradiation, including G1-phase cell-cycle arrest [73]. A rapid recruitment of p21 at the sites of primary DNA damage and the co-localization with other proteins involved in DSB repair (i.e., Mre11, Rad50, and PCNA) were also observed in normal HFs exposed to heavy-ion irradiation [74,75]. The accumulation of p21 to DNA damage foci appeared to occur independently of the assembly of MRN complex (Mre11/Rad50/NBS1), as well as of functional p53. These results were ascribed to an unknown role of p21 in the sensing or early processing of DSBs induced by high LET radiation [75]. Furthermore, p21 was also recruited to local sites in HFs irradiated with a 337 nm laser, conditions producing DSBs, suggesting that the p21 response was independent of the type of lesion and mainly related to PCNA-dependent repair pathways [53]. Accordingly, the accumulation of p21 at laser-irradiated sites was shown to be dependent on PCNA but not on p53 or the NHEJ core factors Ku70, Ku80, and DNA-PKcs [76]. On the other hand, PCNA recruitment to chromatin was observed after the X-irradiation of HFs [77], and a physical interaction with Ku70 and Ku80 heterodimers was found in response to γ-irradiation [78]. These lines of evidence, together with the binding of DNA pol λ to PCNA [79], indicate that PCNA may participate in the NHEJ process. Therefore, the recruitment of p21 to DSB foci appears to be a consequence of its ability to bind PCNA. However, after the exposure of normal foreskin fibroblasts to X-rays, p21 was recruited into nuclear foci spatially distinct from those positive for γ-H2AX and 53BP1, thus suggesting no correlation with DNA DSB repair [80].

These contrasting results can be explained by the different sources of energy employed, producing different types of lesions, including DNA base damage [81]. For the PCNA/p21 interaction to occur, the latter had to be dephosphorylated, as demonstrated in HeLa cells after γ-irradiation [80]. This evidence further supports a PCNA-dependent role of p21, since phosphorylation disrupts the binding to PCNA [82,83]. Nevertheless, the recruitment of PCNA and p21 in NER-deficient XPA fibroblasts, as well as the lack of co-localization with DBS markers, indicated that p21 association with damage sites after ionizing radiation was not related to NER or pathways processing DSBs but to different sub-pathways of BER (in particular, long-patch BER) [80].

### 2.5. Homologous Recombination (HR)

A novel role of p21 has been proposed in the regulation of replication-coupled DSB repair after treatment with DSB- or DNA crosslink-inducing agents (mitomycin C, camptothecin, and etoposide). In fact, p21^−/−^ cells (both human colon cancer cell line HCT116 and MEFs) showed DSB hypersensitivity, which was associated with increased and persistent levels of CDK-mediated BRCA2 S3291 phosphorylation and MR11 nuclear foci formation [84]. These results indicated that in the absence of p21, the NHEJ pathway was activated with a concomitant decrease in HR-dependent repair efficiency. Taken together, these results suggest that p21 may negatively regulate the typical error-prone DSB repair NHEJ, promoting the error-free HR [84]. It is important to note that PCNA is involved in the initiation of DNA pol δ-dependent DNA synthesis associated with recombination [85]. Interestingly, a direct interaction of p21 with the p50 subunit of DNA pol δ was demonstrated in cells treated with adriamycin, a chemotherapeutic drug that induces DSBs [86]. The association between p21 and p50 increased in treated cells, although the influence of this phenomenon on DNA repair was not investigated. The binding between the two proteins occurred in a complex associated with CDK2 but not with PCNA, suggesting an effect related to the cell cycle rather than to DNA repair [86]. However, a mechanism to promote HR by CDK inhibition, as reported above [84], may also be envisaged.

**Table 1 ijms-23-07058-t001:** Roadmap of the discovery of direct interactions of p21 with factors playing a role in DNA repair.

Year	Protein	Activity	DNA Repair System	Refs.
1994	PCNA	DNA synthesis	NER	[31,32,33,34,35,36]
1998	Ku70	damage recognition	NHEJ	[73]
2001	p300	KAT	NER	[56,58,59,60]
2003	PARP-1	synthesis of PAR	BER, SSBR	[66,67,68]
2006	p50 (POLD2)	subunit of DNA pol δ	(not reported)	[84]

### 2.6. Mismatch Repair (MMR)

The MMR system is dedicated essentially to repairing errors generated in the form of mismatches during DNA replication, and it relies on the activity of PCNA necessary to assemble proteins recognizing and processing these lesions. Using different MMR substrates, in vitro studies showed that PCNA is involved not only during the re-synthesis but also in the preceding step of excision, thanks to interactions with MMR-specific factors [87,88,89]. The MMR activity in vitro was inhibited by p21 and reversed by adding PCNA, indicating that the repair was dependent on PCNA and that p21 blocked the reaction by interacting with PCNA [90]. Generally, in this study, the activity of p21 was investigated under excess conditions of p21 with concentrations of p21/PCNA ratio > 3 [90]. Given that in vivo p21 levels during the S phase are very low, the above studies demonstrated the dependence of MMR on PCNA but not that an actual inhibition by p21 may also occur in vivo. Interestingly, after DNA damage induced by alkylating agents, the recruitment of MMR proteins MSH2 and MSH6 to a chromatin-bound fraction was coupled to p21 degradation since a p21 mutant that was resistant to proteolysis inhibited this step of MMR [91].

### 2.7. Translesion DNA Synthesis (TLS)

Translesion DNA synthesis (TLS) represents a fundamental DNA damage tolerance system of cells to bypass DNA lesions that would lead to replication fork stalling in the S phase. This process is activated by PCNA mono-ubiquitination, which triggers a DNA polymerase switch mechanism, replacing the high-fidelity replicative DNA pols with Y-family DNA pols (e.g., pol η, pol κ, pol ι) characterized by poor processivity and low-fidelity [92].

A first study investigating the effect of p21 on PCNA mono-ubiquitination after UV irradiation showed that this modification was inhibited by a stable p21 mutant that could not be degraded because of N-terminal 6xMyc-tag addition. Importantly, the wt protein did not affect the process, and the interaction with CDK, but not with PCNA, was required for the inhibition of PCNA mono-ubiquitination [93]. Another study comparing PCNA mono-ubiquitination levels in cells expressing endogenous p21 and cells treated with siRNA to deplete p21 provided evidence of a positive role of p21 in PCNA modification. However, comparing TLS in p21^+/+^ and p21^−/−^ MEFs, it was shown that p21, through the interaction with PCNA, limited the TLS efficiency, though it provided increased fidelity to the process, thus contributing to keeping the mutagenesis rate of this error-prone system at low levels [94].

In the light of these results, the p21 influence on TLS polymerases was investigated in U2OS cells and in HCT116/p21^+/+^ and p21^−/−^ cells [54]. In particular, the p21 6xMyc stable mutant was shown to interfere with DNA pol η recruitment to PCNA foci after UV irradiation; however, while DNA pol η foci could be detected in p21^+/+^ cells only after UV irradiation, similar foci could be observed also in untreated control p21^−/−^ cells, although to a lesser extent with respect to treated ones. In addition, the number of cells showing pol η foci was higher in p21^−/−^ than in p21^+/+^ cells [54]. The regulatory activity of p21 in the DNA pol switch process after UV irradiation was further investigated using the p21 stable mutant. The results showed that the persistent association of stable p21 with PCNA inhibited the recruitment of DNA pol η, pol κ, pol ι, and Rev1, with the accumulation of DNA replication stress markers [95]. Interestingly, more recent results showed that low levels of p21 are present during the S phase and are required, even in the absence of DNA damage, to prevent DNA pol κ-dependent replicative stress caused by the extremely low fidelity of this enzyme [9].

### 2.8. Fanconi Anemia (FA) Pathway

The FA pathway is activated in response to DNA damage induced by cross-linking agents. In this pathway, FA proteins function together with BRCA1 to repair these lesions [96]. At least one FA protein, such as FANCD2, is known to establish an interaction with PCNA through a PIP box, which then influences the FANCD2 mono-ubiquitination required for DNA repair [97]. Interestingly, in cells lacking p21, FANCD2 mono-ubiquitination is impaired and the protein does not localize to DNA damage foci [98]. The influence of p21 appeared to be exerted at the transcriptional level, through the repression of the de-ubiquitinating enzyme USP1. In the absence of p21, the expression of USP1 resulted in the failure to accumulate the ubiquitinated FANCD2, while a direct intervention of p21 in the interaction among PCNA and FANCD2 to block USP1 de-ubiquitination was not investigated [98]. 

On the basis of the involvement of p21 in the above DNA repair systems, a collective model illustrating the influence of p21 in these processes is reported in Figure 2.

## 3. Influence of p21 Stability in DNA Repair

It has been previously demonstrated that p21 stability is regulated by proteasome-dependent and independent mechanisms, both in the S phase of the cell cycle and after DNA damage [1,2,3]. In particular, the degradation of p21 via the proteasome system requires ubiquitination through different E3 ubiquitin ligases [99,100,101,102,103,104]. The redundancy of the pathways leading to p21 degradation in the S phase has highlighted the importance of maintaining the p21 protein at levels suitable to avoid the block of DNA synthesis. Initially, however, the evidence of p21 degradation after DNA damage suggested the idea that p21 had to be eliminated to promote efficient DNA repair [105]. The causal relationship between p21 levels and DNA repair efficiency was investigated in REF-52 cells by expressing a mutant form in which all six lysine residues were mutated to arginine (K6R), thereby blocking degradation. In such experimental conditions, p21 inhibited NER by preventing the chromatin recruitment of PCNA necessary for DNA repair [105]. In addition, it was shown that the proteasomal degradation of p21 (mediated by the E3 ubiquitin ligase Skp2) was triggered by checkpoint activation and inhibited by caffeine [105]. Further studies investigating UV-induced p21 degradation showed the dependence of this process on CRL4^Cdt2^ and PCNA, using p21 mutants that could not be degraded in human HeLa and HEK293 cells. However, these studies did not analyze the influence of the p21 undegradable mutants on the NER efficiency [102,103]. 

The accumulation of p21 was also induced by CUL4A abrogation in mice and MEFs, since CUL4A is an important component of the CRL4 complex. Surprisingly, the efficiency of CPD removal was increased in *Cul4A*^−/−^, as compared with wt MEFs, but was restored to similar levels when p21 was deleted in *Cul4A*^−/−^ MEFs [6]. These results may be explained by the concomitant increase in DDB2 and XPC proteins (also under the control of CUL4A), which could activate the GGR pathway, typically absent in rodent cells [106]. It is worth noting that in this model system, the accumulation of endogenous p21 provided a positive effect on NER and concomitantly avoided the occurrence of mitotic defects observed in *p21*^−/−^ and in *Cul4A*^−/−^/*p21*^−/−^ MEFs after UV-induced DNA damage [6].

An important factor involved in the signaling of p21 degradation is the post-translational modification (PTM) via phosphorylation [83,105]. This mechanism was supported by p21 phosphorylation mediated by LKB1 and by the downstream kinase NUAK1, whose downregulation and depletion promoted p21 accumulation after UVB irradiation [107]. An LKB1-deficient mouse model showed that these kinases are able to bind directly and phosphorylate p21 (at T80 and T146), and provided a direct mechanism connecting LKB1-dependent p21 phosphorylation with its degradation. The overexpression of LKB1^wt^ in HaCaT keratinocytes favored DNA repair compared with a kinase-dead mutant. In addition, the depletion of LKB1 induced, together with p21 accumulation, reduced efficiency in the removal of UVB-induced lesions (both CPD and 6,4-PP) and resistance to apoptosis. The concomitant depletion of LKB1 and p21 relieved the decrease in CPD removal, showing that DNA repair efficiency was improved by reducing the amount of p21 accumulated by LKB1 deficiency [107]. These findings support the idea that the fine regulation of p21 levels is required for optimal DNA repair.

Another example of reduced DNA repair, particularly PCNA-dependent DNA repair induced by high p21 levels (due to impaired degradation), was provided by p53-deficient cell models in which the persistent expression of p21 induced the escape from senescence, enabling cell proliferation in the presence of high p21 levels [18]. In these model systems, p21 accumulation was ascribed to the saturation of CRL4 ubiquitin ligase, which blocked p21 degradation, with the consequent impairment of DNA repair systems, such as TLS, BER, as well as NER [108]. In the TLS process, p21 induced a decrease in the amount of monoubiquitinated PCNA and chromatin-bound DNA pol η. In contrast, BER and NER were affected at the transcriptional level by down-regulating the expression of key factors, such as DNA glycosylases, APE1 and Lig3 (BER), and DDB1, XPC, XPG, and XPF (NER), thereby fueling genomic instability [108].

However, when elevated p21 protein levels were induced in primary HF by HDAC inhibitors, such as trichostatin A, the amount of p21 recruited to UV-induced DNA damage sites increased in concomitance with unscheduled DNA synthesis (UDS), indicating that under these conditions, p21 did not inhibit NER [53]. Using a different approach, caffeine was used to inhibit the checkpoint activating the degradation pathway [105] of endogenous p21 in HF cells. Even in these conditions, however, the efficiency of NER, as assessed by UDS, was only slightly affected [109]. The difference between this and the previously mentioned studies may rely on the amount of endogenous p21 protein vs. that produced by the ectopic expression of an undegradable form [93,105]. In particular, the levels reached by the mutant form resistant to degradation may have blocked DNA repair by saturating the binding of p21 to PCNA, with the possible displacement of PCNA partners [24]. In contrast, the levels of endogenous p21 available after blocking the degradation with caffeine were probably not sufficient to saturate PCNA. Consequently, the binding of partners (e.g., the p125 subunit of DNA pol δ) was not disrupted, and UDS was only modestly reduced [109]. These studies indicate that the inhibition of DNA repair by p21 is a matter of the overproduction and excess accumulation of the protein and highlight the importance of regulating the amount of p21 available, which may produce toxic effects when degradation is impaired [110]. 

## 4. Protein Degradation after DNA Damage: A DNA Repair-Coupled Mechanism

The proteolytic destruction of DNA repair factors is a process known for many years, although the exact role has remained, for some time, not well understood [111]. Only recently has the degradation of DNA replication and repair proteins been analyzed in more detail, and various lines of evidence now indicate that this process is necessary to remove proteins from chromatin after they have completed their function [112,113,114]. This requirement plays an important role in genome stability, since the chromatin retention of DNA repair factors is known to activate a DNA damage response and block DNA repair [115,116]. Therefore, their removal is necessary in order to avoid a dangerous permanence at DNA repair sites. In agreement with this view, it has been very recently shown that the segregase p97/VCP/Cdc48 may carry out this function during DNA repair by extracting ubiquitinated proteins from the chromatin compartment [117]. The chromatin retention of NER factors such as DDB2 and XPC occurred when p97 was expressed as an inactive mutant form or down-regulated by RNAi. The prolonged presence of such factors causes genotoxicity, e.g., with the production of chromosomal rearrangements [117], thus highlighting the need to remove DNA repair factors from chromatin once they have performed their task. Interestingly, the mechanism of the chromatin extraction of NER factors via p97 is linked to their ubiquitination mediated by CUL4A [118]. 

The CRL4 complex is composed of the adaptor protein DDB1 and a substrate receptor DCAF (DDB1 and CUL4-associated factor), such as Cdt2. The CRL4^Cdt2^ complex also utilizes PCNA as a docking platform for some substrates that bind DNA [119]. The recognition of these protein substrates by PCNA is mediated by a specialized PIP box, termed PIP degron, which facilitates the degradation, as compared with other PCNA partners [120,121].

Considering that the ubiquitination of p21 is the pre-requisite for its degradation after DNA damage, the interaction with PCNA and the CRL4 complex appears determinant for successful DNA repair. In fact, the depletion of the components of the CRL4 complex, such as DDB1 or Cdt2, has been shown to impair the NER efficiency, although the depletion or ablation of CUL4A provided a different response [6,122,123]. During NER, a CRL4 complex containing DDB2 in substitution of Cdt2 is formed, and CRL4^DDB2^ promotes the ubiquitination of XPC [124]. Remarkably, DDB2 depletion produces an increase in p21 levels, suggesting that their interaction is required for subsequent p21 degradation [51]. However, the interaction with PCNA and not with DDB2 is important for p21 degradation, since DDB2 may associate with p21 only in the presence of PCNA [125], and a NER-deficient DDB2 mutant unable to interact with PCNA does not bind p21 [126]. These findings indicate that p21 degradation is promoted by multiple CRL4 complexes containing different DCAFs, whose specific choice is, however, still unclear.

Since the mechanism of degradation appears to be shared with core DNA repair factors, we have examined the literature and found that after DNA damage (mainly after UV radiation), other PCNA interactors participating in DNA repair are degraded via the ubiquitin–proteasome pathway. In the NER system, in addition to DDB2, a notable example is provided by XPG, which has been found to be degraded after DNA damage induced by UV and cisplatin through the interaction with Cdt2 [123]. Interestingly, XPG degradation facilitated the recruitment of the p125 subunit of DNA pol δ required for gap-filling DNA synthesis, i.e., the next step after lesion incision/excision, thus suggesting a DNA repair-coupled mechanism of degradation [123].

Another protein containing a PIP-box, p12, i.e., the fourth subunit of DNA pol δ (POLD4), was shown to be degraded after DNA damage [127]. In a way similar to p21, p12 degradation was dependent on checkpoint activation by the ATR kinase and mediated by the CRL4^Cdt2^ complex during the S phase [128]. The significance of p12 degradation in DNA repair is not yet fully elucidated, although it seems required to switch the composition of DNA pol δ from four to three components [127]. Further studies showed that at least two E3 ligases are involved in p12 degradation after DNA damage, i.e., RFN8 and CRL4^Cdt2^ [129,130].

Several examples of the degradation of other DNA repair factors are available, highlighting this process as a regulatory mechanism involved in the DNA damage response. Some of these factors undergo ubiquitination and proteasomal destruction by the CUL4-dependent pathway, such as the Rev7 subunit of DNA polymerase ζ, a member of the Y-family DNA pols, which is also involved in TLS in response to UV-induced DNA damage [131]. The DNA helicase FBH1 is involved in HR and accumulates at sites of DNA damage in a PCNA-dependent manner [132]. The interaction with PCNA was required for FBH1 degradation mediated by the CRL4 complex through a PIP-degron motif: the mutation of this sequence, impeding the degradation, resulted in an impairment of DNA pol η recruitment after UV-induced damage [132]. Similarly, the acetyltransferase HBO1 was found to be targeted by the CRL4^DDB2^ complex for degradation after UV irradiation, with an ATM/ATR-mediated phosphorylation mechanism [133]. HBO1 was later reported to play a significant role in NER, thus connecting DNA repair with the degradation process [134]. Furthermore, DNA ligase I was targeted for ubiquitination by the CUL4/DDB1/DCAF7 complex, both after the inhibition of DNA replication and after DNA damage [135].

In the *Xenopus* model system, Thymine DNA Glycosylase (TDG), an enzyme acting in the first step of BER, in transcription as well as in DNA methylation, was identified as another substrate of the CRL4^Cdt2^ complex because of the interaction with PCNA and the presence of a conserved degron motif in its sequence [136]. Interestingly, TDG degradation was also found in human cells and shown to be important to prevent toxic effects from an excess of this protein [137].

Other E3 ligases, however, are also involved in the regulatory ubiquitination/proteasomal removal of DNA repair factors, such as, for instance, the degradation of DNA pol η, in which two different E3 ubiquitin ligases have been identified [138,139]. The DNA damage-induced degradation of the exonuclease Exo1, mediated by the Skp1 complex, was shown to couple the activity of this enzyme with a mechanism limiting excessive DNA end resection to promote correct DSB repair [140]. 

Interestingly, the PIP degron motif, and not simply the PIP box, was indicated as the mediator in the switch of PCNA partners to facilitate the recruitment of TLS polymerases at DNA damage sites, thus supporting the coupling of CRL4 activity with DNA repair [141].

These studies indicate that the proteasomal degradation of relevant factors during DNA repair is an important mechanism of genome stability maintenance which is required, similarly to the S-phase-coupled process, to avoid excessive accumulation and to limit the activity of such factors. From this point of view, sharing the same mechanism of degradation, p21 appears to behave as a DNA repair regulator.

## 5. p21 Fine Regulation

The major reason responsible for the apparent contrasting effects induced by p21 on different DNA repair systems is undoubtfully the amount of protein present or produced in the various experimental settings and systems. Excess p21 protein in vitro, or expressed in vivo, particularly as an undegradable mutant, may have reached saturating effects, especially on the association with PCNA, which may have easily blocked DNA repair reactions [105,108]. 

These observations explain the need for a fine regulation of p21 levels, as recently emphasized by new findings showing that low amounts of p21 remaining during the S phase are required to keep the replication fork speed within “normal” levels [9,10]. The depletion of p21 promoted an increase in the fork speed, causing replication stress [10]. In addition, p21 was suggested to regulate the polymerase choice, preventing the accumulation of genome instability caused by excessive replication defects introduced by alternative polymerases, such as DNA pol k [9,142]. This mechanism is supported by the p21 degradation kinetics, which are apparently delayed compared with that of the replication licensing factor Cdt1 [142]. An anticipated removal of p21 caused difficulties in maintaining active DNA replication, and p21 depletion induced disturbances of nucleotide incorporation, culminating in the failure of the S phase [143]. In agreement with these findings, previous works showed that p21 could protect arrested replication forks from collapse after topoisomerase I inhibition [144] and that p21 limited the extent of DNA damage in the S phase induced by concomitant Wee1 inhibition and ionizing radiation [145]. These results indicate that the amount of cellular p21 must be finely regulated to low levels suitable to maintain a correct DNA replication rate, ensuring a safe progression through the S phase [142,143]. A similar mechanism of fine regulation could be then useful for DNA repair synthesis.

## 6. Conclusions and Perspectives

The evidence that the orderly degradation of p21, in a way similar to the S phase, also occurs after DNA damage [143] suggests that p21 regulates DNA repair and that this activity must be performed before its degradation [142]. In particular, the interaction with PCNA might be necessary for regulatory purposes and not only instrumental to the substrate recognition by the CRL4 complex. This hypothesis, in addition to previous studies indicating deficiencies in DNA repair in the absence of p21 [37,38,39,44], is supported by the evidence that p21/PCNA interaction is required to prevent chromosomal aberrations after DNA damage, which was observed in cells expressing a p21 mutant unable to interact with PCNA [110].

A possible regulatory role for p21 could be the switching of DNA repair factors interacting with PCNA, as suggested for TLS polymerases [141]. This idea is supported by the evidence that p21 was found in a complex with PCNA and DNA pol δ, but also with XPG in the chromatin fraction of UV-irradiated, quiescent fibroblasts [53,58]. These associations would not be justified if p21 were to interact with PCNA exclusively for degradation. In agreement with this role, XPG has been shown to abnormally persist on chromatin after UV irradiation in cells lacking p21 [59]. The switching mechanism could be coupled to degradation and would serve to coordinate and limit the activity of DNA repair factors, as observed for DNA pol η in TLS [94], for PARP1 in BER [67], and for DDB2, XPC, and XPG in NER [116,117,123], thereby avoiding a prolonged presence at DNA damage sites. 

A model depicting how p21 may regulate PCNA partners coupled with degradation during NER is schematically shown in Figure 3. The first step in which PCNA is present during NER is DNA incision, where PCNA may stimulate the activity of XPF [146], preceding that of XPG, which is known to interact with PCNA [147]. In this step, p21 may facilitate the release of XPG by coupling its degradation via the CRL4 complex (Figure 3A,B) in order to promote the transition from incision to the DNA synthesis step [123]. This picture is suggested by the persistence of XPG, as well as PCNA, at DNA damage sites in p21^−/−^ cells [44,59]. DNA pol δ is assembled with PCNA, and after DNA synthesis, p21 binding may promote pol δ subunit dissociation with consequent p12 and p21 ubiquitination by the CRL4 complex for degradation [102,103,128,130]. In the absence of p21, the catalytic subunit p125 is accumulated (our unpublished results) in a way similar to XPG [59], thus suggesting a mechanistic role for p21 in the PCNA partner switch/degradation (Figure 3C,D). 

Finally, it is worth noting that p21 degradation is instrumental to the induction of the apoptotic program when cells are challenged with high levels of DNA damage that may be difficult to repair. In fact, p21 is a known inhibitor of apoptosis, and its degradation has been shown to facilitate the elimination of these cells, as pointed out in previous works [109,148,149,150]. The evidence that synthetic lethality induced by DNA damaging agents and checkpoint inhibitors may be attained in combination with p21 loss has provided further indications on the role of p21 in protecting against apoptosis induced by DNA damage [151]. 

In the future, the idea that p21 is required to limit DNA repair synthesis in space and time, and that this activity is coupled to degradation, should be tested experimentally in order to clarify whether and when p21 may be beneficial in the response to therapeutic approaches with DNA-damaging drugs. 

## Figures and Tables

**Figure 1 ijms-23-07058-f001:**
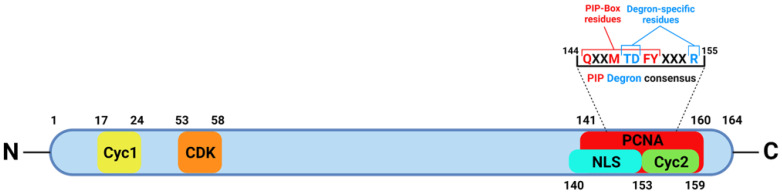
Schematic representation of p21’s structure, showing regions involved in the interactions with cyclins (Cyc1 and 2), CDK, and PCNA. The sequence of the PIP box and specific degron residues are also indicated. NLS—nuclear localization signal. Created with https://biorender.com/ (Toronto, ON, Canada) (accessed on 10 May 2022).

**Figure 2 ijms-23-07058-f002:**
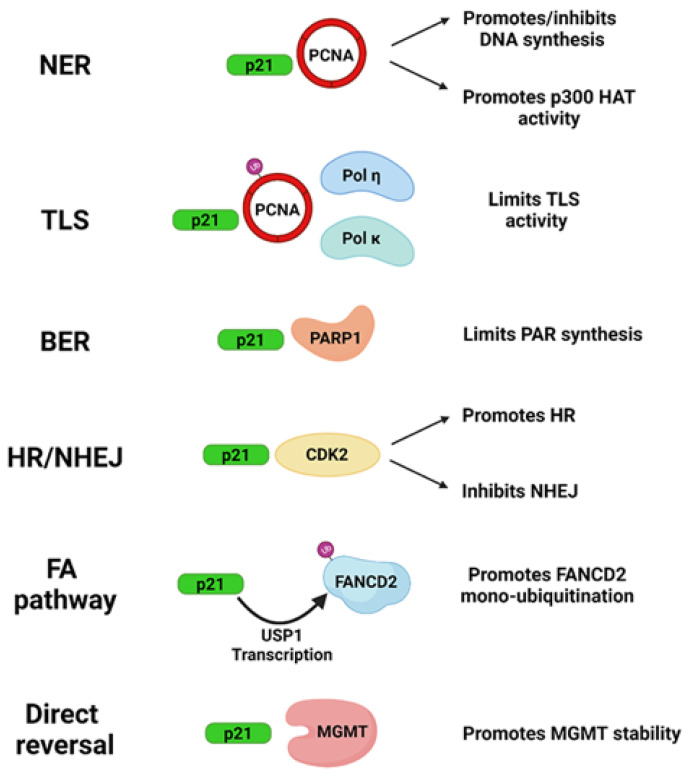
Schematic model summarizing the influence of p21 on DNA repair systems. Created with https://biorender.com/ (Toronto, ON, Canada) (accessed on 10 May 2022).

**Figure 3 ijms-23-07058-f003:**
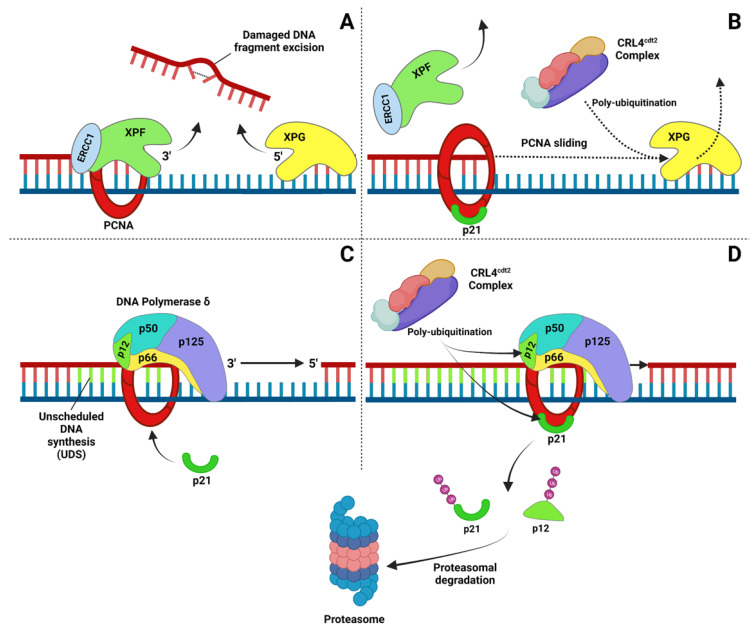
Schematic representation of p21 function during DNA repair. The example shown here is pertaining to NER, but it could be applied to other repair mechanisms where PCNA is involved. (**A**,**B**) The first step requiring PCNA in NER involves the incision of the fragment by XPF (stimulated by PCNA) and by XPG. The transition through this step could be favored by p21 enabling the release of XPG (and perhaps XPF) from PCNA for subsequent degradation (dotted line) to allow the next step of DNA pol δ-dependent DNA synthesis. (**C**,**D**) Similar to the previous step, p21, before being itself ubiquitinated and degraded, may be required to promote p12 subunit removal for degradation. Created with https://biorender.com/ (Toronto, ON, Canada) (accessed on 10 May 2022).

## Data Availability

Not applicable.

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
