# Peer review of "Revisiting the Function of p21CDKN1A in DNA Repair: The Influence of Protein Interactions and Stability"

_ijms, 2022, doi:10.3390/ijms23137058_

Round 1

Reviewer 1 Report

The review "Revisiting the function of p21CDKN1A in DNA repair: influence of protein interactions and stability" submitted here is an interesting topic.  

I have some concerns regarding the language and sentences are very confusing. Authors should take help of professional language services. I would like to point out some of my observations below:

1. line no. 14 cell motility, thanks to interactions with specific proteins

Authors need to provide reference regarding mobility and thanks to protein is not a scientific and professional language.

2. Line no. 88 to 89 The participation of p21 in DNA repair processes has been initially investigated not because of its activity of cell cycle inhibitor, necessary for giving the cell time to remove the damage, but mainly because of its ability to inhibit DNA synthesis thanks to the high-affinity binding to PCNA.

3. Line no. 92 to 94, These processes include nucleotide excision repair (NER), base excision repair (BER), mis- match repair (MMR), as well as homologous recombination (HR) and non-homolo- gous end joining (NHEJ). Therefore, whether p21 could be an inhibitor of DNA repair became a question to address, causing a longstanding debate.

(what evidences you have role of p21 in NER, BER, MMR, NHEJ and HR? Please provide references individually.

3. Line no 95, Authors are confused with their statement: , whether p21 could be an inhibitor of DNA repair became a question to address, causing a longstanding debate.

Overall, I would recommend to rewrite the manuscript and propose a collective model with adequate references.

4. Provide a roadmap of discovery regarding specific proteins.

Author Response

We thank this Reviewer for his comments and helpful criticisms.

Points 1-3. We understand and regret that the manuscript language was confused. The language has been now revised by a native language English scientist, and rewriting, where appropriated, has been done (modified sentences appear in red). In particular, all the sentences related to points 1-3, have been rewritten.

  1. The reference for p21 mobility (migration) was already present (ref. 11) in the Introduction. In the revised Ms, it is at line 45 p. 2. The first appearance is in the Abstract, where usually no references are given.
  2. This sentence has been rewritten.
  3. This sentence made reference to the DNA repair processes in which PCNA (and not p21) is involved. We are sorry for the misunderstanding. Of course, the sentence has been rewritten.

We are grateful for the suggestion of a collective model of the role of p21 in DNA repair. We have prepared a new figure (no. 2) that recapitulates the functions of p21 in each repair system, that have been treated in paragraphs 2.1-2.8, where individual references are presented. We have preferred not to include references in the Figure itself, to avoid a too crowded picture.

  1. The Table 1 has been modified in order to provide a roadmap of discovery of specific interactions.

Reviewer 2 Report

The review article "Revisiting the function of p21 in DNA repair: Influence of protein interactions and stability" presented by Ticli et al. is a very well written review about state of the art of p21 involvement in DNA repair and is presenting very new aspects about p21 regulation of DNA repair as well as its impact on DNA replication. The article is based on the state of knowledge concerning cell cycle arrest, apoptosis etc. and leads from there to various DNA repair processes and p21 interaction during these ones. With 150 citations it gives a comprehensive overview of many aspects in repair control. I recommend the publication of this article. There is only a minor suggestion which I have and which the authors may consider in the final version of the text.

Many of the repair processes described are usually DNA repair processes after UV-irradiation which is mentioned at several text paragraphs. However, it may be clearified that processes like NHEJ or HRR are processes that also occur after x-ray, gamma-ray, or particle irradiation. So it may be helpful for the reader if the authors could always mention the DNA damage sources for which the mentioned repair processes and p21 interaction mainly function.

Author Response

We thank this Reviewer for the nice comments and suggestions.

It is true that often in the manuscript, the works described have used UV damage. However, in the paragraphs related to NHEJ and to HR, the source of damaging agents (in the form of X, gamma rays, or LET), was already reported (see par. 2.4 and 2.5). If any other description has been inadvertently omitted, we’ll be glad to add to the manuscript.

Round 2

Reviewer 1 Report

Authors have worked on my concerns. I am in favor of publication.